# Towards simple time-to-event modeling: optimizing neural networks via rank regression

## Abstract

Time-to-event analysis, also known as survival analysis, aims to predict the first occurred event time, conditional on a set of features. However, the presence of censorship brings much complexity in learning algorithms due to data incompleteness. Hazard-based models (e.g. Cox's proportional hazards) and accelerated failure time (AFT) models are two popular tools in time-to-event modeling, requiring the proportional hazards and linearity assumptions, respectively. In addition, AFT models require pre-specified parametric distributional assumptions in most cases. To alleviate such strict assumptions and improve predictive performance, there have been many deep learning approaches for hazard-based models in recent years. However, compared to hazard-based methods, AFT-based representation learning has received limited attention in neural network literature, despite its model simplicity and interpretability. In this work, we introduce a Deep AFT Rank-regression for Time-to-event prediction model (*DART*), which is a deep learning-based semiparametric AFT model, and propose a $l_1$-type rank loss function that is more suitable for optimizing neural networks. Unlike existing neural network-based AFT models, the proposed model is semiparametric in that any distributional assumption is not imposed for the survival time distribution without requiring further hyperparameters or complicated model architectures. We verify the usefulness of *DART* via quantitative analysis upon various benchmark datasets. The results show that our method has considerable potential to model high-throughput censored time-to-event data.

## 1 Introduction

Time-to-event analysis, also known as survival or failure time analysis, is a major statistical approach in various fields such as biostatistics, medicine, and economics to estimate either risk scores or the distribution of event time, given a set of features of subjects (Viganò et al., 2000; Cheng et al., 2016; Dirick et al., 2017; Li et al., 2021). There are benefits of assessing risk or quantifying survival probabilities but, for all that, time-to-event analysis itself is challenging because of the existence of censoring. In real-world studies, a subject (e.g. a patient in medical research) can drop out before events of interest (e.g. death) happen, so that one can not follow them up (Leung et al., 1997). The presence of censoring in survival data can create a serious challenge in applying standard statistical modeling strategies. Usually, the censoring process is assumed to be non-informative in that it is irrelevant of the underlying failure process given features, but should be properly accounted for, otherwise leading to biased results.

The most popular and standard approach for modeling time-to-event data is to use Cox's proportional hazards (CoxPH) model. CoxPH relates a conditional hazard to given features in a multiplicative form between the baseline hazard function and exponentiated regression component, and consequently learns relative risks. It often works on the assumptions of proportional hazards and time-invariant covariate-effects, which are difficult to follow in the real world (Aalen, 1994). Statistical testing procedures, such as Schoenfeld's test, are usually conducted to examine these underlying assumptions since many Cox-based analyses are vulnerable to violation of model assumptions. (Aalen & Gjessing, 2001; Kleinbaum & Klein, 2010).

The accelerated failure time model (AFT) or accelerated life model relates the logarithm of the failure time linearly to the features. As a result of its direct physical interpretation and the connection

with linear models, this model provides an attractive alternative to the CoxPH for the regression analysis of censored failure time data. Unlike CoxPH, standard AFT model parametrizes the underlying time-to-event distribution up to a set of finite-dimensional parameters such as Weibull and log-normal (Lee & Wang, 2003). However, imposing distributional assumption is too strict in real data analysis and can reduce the attractiveness inherent in the AFT model, mostly underperforming Cox-based analysis (Cox, 2008; Kleinbaum & Klein, 2010). Recently, based on statistical theories and the advent of deep learning techniques, various time-to-event models have been explored to circumvent the necessity of assumptions such as linearity, single risk, discrete time, and fixed-time effect (Katzman et al., 2018; Lee et al., 2018; Ren et al., 2019; Kvamme & Borgan, 2019; Avati et al., 2020; Tarkhan et al., 2021; Rahman et al., 2021).

For example, *Cox-Time* (Kvamme et al., 2019) and *DATE* (Chapfuwa et al., 2018) alleviate the most fundamental but strict assumptions of the CoxPH and parametric AFT models by achieving non-proportional hazards and non-parametric event-time distribution, respectively. *Cox-Time* exploits the neural network as a relative risk function to model interactions between time and covariates. The authors also show that the proposed loss function is a good approximation for the Cox partial log-likelihood. *DATE* is a conditional generative adversarial network for implicitly specifying a time-to-event distribution of ATF model. It does not require the pre-specified distribution in parametric form, instead, the generator can learn it from the data with the adversarial loss function. Incidentally, various deep learning-based approaches have been spotlighted to improve performance by resolving issues such as temporal dynamics and calibration (Lee et al., 2019; Nagpal et al., 2021; Gao & Cui, 2021; Kamran & Wiens, 2021; Hu et al., 2021). Therefore, it became important to utilize well-designed objective functions that fit not only statistical backgrounds but also optimization of neural networks.

In this paper, we introduce a Deep AFT Rank-regression for Time-to-event prediction model (*DART*), a deep learning-based semiparametric AFT model trained with an objective function originated from Gehan's rank statistic. The model does not require specifying event time distribution while keeping the advantage of the standard AFT model that directly predicts event time. With a simple form of the loss function, by constructing comparable rank pairs, the optimization of *DART* is efficient compared to other deep learning-based time-to-event models. Experimental results show that *DART* is not only well-calibrated but also competitive in event order prediction performance even compared to hazard-based models. Furthermore, we believe that this work can be widely applied in the community while giving prominence to advantages of the AFT model that is relatively unexplored.

## 2 RELATED WORKS

We first overview time-to-event modeling focusing on the loss functions of *Cox-Time* and *DATE* models to highlight the difference in concepts before introducing our method. The primary interest of time-to-event analysis is to estimate survival quantities like survival function $S(t) = P(T \geq t)$ or hazard function $h(t) = \lim_{\delta \to 0} P(t \leq T \leq t + \delta | T \geq t)/\delta$, where $T \in \mathbb{R}^+$ denotes time-to-event random variable. In most cases, due to censored observations, those quantities cannot be directly estimated with standard statistical inference procedure. In the presence of right censoring, Kaplan & Meier (1958) and Aalen (1978) provided consistent nonparametric survival function estimators, exploiting right-censoring time random variable $C \in \mathbb{R}^+$. Researchers then can get stable estimates for survival quantities with data tuples $\{y_i, \delta_i, X_i\}_{i=1}^N$, where $y_i = \min(T_i, C_i)$ is the observed event time with censoring, $\delta_i = I(T_i \leq C_i)$ is the censoring indicator, and a vector of features $X_i \in \mathbb{R}^P$. Here, $N$ and $P$ denote the number of instances and the number of features, respectively. While those nonparametric methods are useful, one can improve predictive power by incorporating feature information in a way of regression modeling. Cox proportional-hazards (CoxPH) and accelerated-failure-time (AFT) frameworks are the most common approaches in modeling survival quantities utilizing both censoring and features.

### 2.1 HAZARD-BASED MODELS

A standard CoxPH regression model (Cox, 1972) formulates the conditional hazard function as:

$$h(t|X_i) = h_0(t) \exp(\beta^T X_i), \ (i = 1, \dots, N), \tag{1}$$

where $h_0(\cdot)$ is an unknown baseline hazard function which has to be estimated nonparametrically, and $\beta \in \mathbb{R}^P$ is the regression coefficient vector. It is one of the most celebrated models in statistics in that $\beta$ can be estimated at full statistical efficiency while achieving nonparametric flexibility on $h_0$ under the proportionality assumption. Note the model is semiparametric due to the unspecified underlying baseline hazard function $h_0$. Letting $\mathcal{R}_i$ be the set of all individuals "at risk", meaning that are not censored and have not experienced the event before $T_i$, statistically efficient estimator for regression coefficients can be obtained minimizing the loss function with respect to $\beta$:

$$L_{\text{CoxPH}}(\beta) = \sum_i \delta_i \log \left( \sum_{j \in \mathcal{R}_i} \exp \left[ \beta^T X_j - \beta^T X_i \right] \right), \tag{2}$$

which is equivalent to the negative partial log-likelihood function of CoxPH model.

Based on this loss function, Kvamme et al. (2019) proposed a deep-learning algorithm for the hazard-based predictive model, namely *Cox-Time*, replacing $\beta^T X_j$ and $\beta^T X_i$ with $g(y_j, X_j; \theta)$ and $g(y_i, X_i; \theta)$, respectively. Here, $g(\cdot)$ denotes the neural networks parameterized by $\theta$, and $\mathcal{R}_i$ would be replaced by $\tilde{\mathcal{R}}_i$, representing the sampled subset of $\mathcal{R}_i$. With a simple modification of the standard loss function in Eq. (2), *Cox-Time* can alleviate the proportionality for relative risk, showing empirically remarkable performance against other hazard-based models in both event ordering and survival calibration.

## 2.2 ACCELERATED-FAILURE-TIME MODELS

The conventional AFT model relates the log-transformed survival time to a set of features in a linear form:

$$\log T_i = \beta^T X_i + \epsilon_i, \ (i = 1, \ldots, N), \tag{3}$$

where $\epsilon_i$ is an independent and identically distributed error term with a common distribution function $F_0(\cdot)$ that is often assumed to be Weibull, exponential, log-normal, etc. As implied in Eq. (3), AFT model takes a form of linear modeling and provides an intuitive and physical interpretation on event time without detouring via the vague concept of hazard function, making it a powerful alternative to hazard-based analysis. However, imposing a parametric distributional assumption for $\epsilon_i$ is a critical drawback of the model, for which model in Eq. (3) could be a subclass of the hazard-based models.

To alleviate linearity and parametric distributional assumptions, several works brought the concept of generative process and approximated the error distribution via neural networks like generative adversarial network (GAN) (Miscouridou et al., 2018; Chapfuwa et al., 2018). Especially, Chapfuwa et al. (2018) proposed a deep adversarial time-to-event (*DATE*) model, which specifies the loss function as:

$$\begin{aligned}
L_{\text{DATE}}(\theta, \phi) = \ &\mathbb{E}_{(X,y) \sim F_{nc}}[D_\phi(X, y)] + \mathbb{E}_{X \sim F_{nc}, \xi \sim F_\xi}[1 - D_\phi(X, G_\theta(X, \xi; \delta = 1))] \\
&+ \lambda_2 \mathbb{E}_{(X,y) \sim F_c, \xi \sim F_\xi}[\max(0, y - G_\theta(X, \xi; \delta = 0))] \\
&+ \lambda_3 \mathbb{E}_{(X,y) \sim F_{nc}}[\|t - G_\theta(X, \xi; \delta = 1)\|_1]
\end{aligned} \tag{4}$$

where $\theta, \phi$ denotes the parameter set associated with a generator $G_\theta$ and a discriminator $D_\phi$, respectively, $(\lambda_2, \lambda_3)$ are hyperparameters to tune censoring trade-off, $F_{nc}(X, y)$ and $F_c(X, y)$ are empirical joint distributions for non-censored cases and censored cases, respectively, and $F_\xi$ is the simple distribution, such as uniform distribution. The generator $G_\theta$ implicitly defines event time distribution. Despite *DATE* achieves prominent survival calibration via the sample-generating process, the objective function is quite complicated and the GAN framework is inherently prone to mode collapse, i.e., the generator learns only a few modes of the true distribution while missing other modes (Srivastava et al., 2017). Also, when optimizing neural networks with multiple loss functions, it is difficult to balance and there might be conflicts (i.e. trade-off) with each term (Dosovitskiy & Djolonga, 2020). Therefore, their loss function might be difficult to be optimized as intended and requires a burdening training time, and consequently not be suitable for large-scale time-to-event analysis.

In the statistical literature, there have been many attempts to directly estimate regression coefficients in the semiparametric AFT model, where the error distribution $F_0$ is left unknown, rather than imposing specific parametric distribution or exploiting generative models. In this work, we bridge

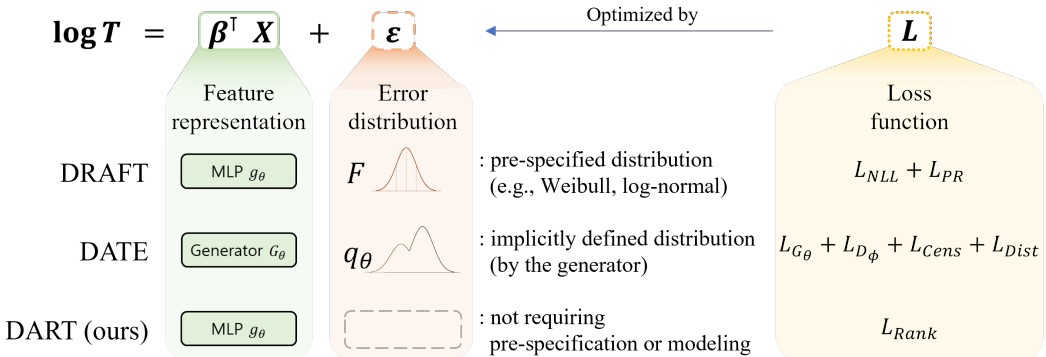

Figure 1: Illustration of conceptual differences between deep learning-based AFT models in terms of their respective contributions and required assumptions with a format of the standard AFT. To alleviate the parametric distribution assumption, which *DRAFT* has, *DATE* exploits the GAN framework and learns the implicit underlying distribution $q_\theta$ through the generator parameterized by $\theta$. For *DRAFT*, $L_{NLL}$ and $L_{PR}$ denote negative log-likelihood and partial ranking likelihood, respectively. *DATE* basically requires four loss functions: $L_{G_\theta}, L_{D_\phi}$ for the generator and the discriminator, $L_{Cens}$ for adjusting censoring distribution, and $L_{Dist}$ for the distortion penalty. Compared to the others, *DART* does not require pre-specification or modeling for error distribution and it is trained with a simple loss function supported by statistical theory.

non-linear representation learning and an objective function for estimation of semiparametric AFT model, which is originated from Gehan's rank statistic. By extensive quantitative analysis, we have shown the beauty of simplicity and compatibility of rank-based estimation, along with outstanding experimental performance.

## 3 METHOD

In this section, we introduce the concept of *DART*, followed by predictive analysis for survival quantities. The conceptual differences with the other neural network-based AFT models are illustrated in Figure 1. The semiparametric AFT is distinct from a parametric version in that the error distribution function $F_0$ is left completely unknown like the baseline hazard function in the CoxPH. By further exploiting neural networks, we propose *DART* model that can be formulated as a generalization of model in Eq. (3):

$$\log T_i = g(X_i; \theta) + \epsilon_i, \ (i = 1, \dots, N), \tag{5}$$

where $g(X_i; \theta)$ denotes arbitrary neural networks with input feature vector $X_i$ and a parameter set $\theta$, outputting single scalar value as predicted log-scaled time-to-event variable. With this simple and straightforward modeling, *DART* entails several attractive characteristics over existing AFT-based models. First, the semiparametric nature of *DART* enables flexible estimation of error distribution, allowing improved survival prediction via neural network algorithms for $F_0$. Second, the restrictive log-linearity assumption of AFT model can be further alleviated by exploiting deep neural networks. Specifically, while standard AFT model relates time-to-event variable to feature variable in linear manner, deep learning is able to approximate any underlying functional relationship, lessening linearity restriction. Although *DART* still requires log-transformed time as a target variable, its deep neural network redeems the point with powerful representative performance supported by universal approximation theorems, enabling automated non-linear feature transformation (Leshno et al., 1993; Schäfer & Zimmermann, 2006; Zhou, 2020).

### 3.1 PARAMETER ESTIMATION VIA RANK-BASED LOSS FUNCTION

In statistical literature, many different estimating techniques have been proposed for fitting semiparametric AFT model (Tsiatis, 1990; Jin et al., 2003; 2006; Zeng & Lin, 2007). Among them, we shall adopt the $l_1$-type rank-based loss function by taking into account the censoring information, which is efficient and suitable for stably fitting neural networks. We also describe two

alternative optimization methods in the appendix as extensions of *DART*. Letting a residual term $e_i \equiv e_i(\theta) = \log y_i - g(X_i; \theta)$, the objective loss function for *DART* can be formulated as:

$$L_{\text{Rank}}(\theta) = \frac{1}{N} \sum_{i=1}^{N} \sum_{j=1}^{N} \delta_i(e_i - e_j) I\{e_i \geq e_j\}, \tag{6}$$

where $I(\cdot)$ is the indicator function that has value 1 when the condition is satisfied, otherwise 0. The estimator $\hat{\theta}$ can be obtained by minimizing the loss function with respect to model parameter set $\theta$. Optimization of model parameters can be conveniently conducted via batched stochastic gradient descent (SGD). Notice that the loss function in Eq. (6) involves model parameter $\theta$ only, without concerning estimation of the functional parameter $F_0$, enabling simple time-to-event regression modeling.

Strength of the loss function is theoretical consistency of optimization without requiring any additional settings. Let the neural network be expressed: $g(X_i; \phi, \beta) = \beta^T W_i$, where $W_i \in \mathbb{R}^K$ is transformed feature vector through hidden layers with parameter set $\phi$, and $\beta \in \mathbb{R}^K$ is a parameter set of linear output layer. Then, it is easy to see that the following estimating function is the negative gradient of the loss function with respect to $\beta$:

$$U_{\text{Rank}}(\beta) = \frac{1}{N} \sum_{i=1}^{N} \sum_{j=1}^{N} \delta_i(W_i - W_j) I(\log y_i - \beta^T W_i \leq \log y_j - \beta^T W_j) \overset{\text{set}}{=} 0. \tag{7}$$

Eq. (7) is often called the form of Gehan's rank statistic (Jin et al., 2003), testing whether $\beta$ is equal to true regression coefficients for linear model $\log T_i = \beta^T W_i + \epsilon_i$, and the solution to the estimating equation $\hat{\beta}$ is equivalent to the minimizer of Eq. (6) with respect to $\beta$. This procedure entails nice asymptotic results such as $\sqrt{n}$-consistency and asymptotic normality of $\hat{\beta}$ under the counting processes logic, assuring convergence of $\hat{\beta}$ towards true parameter $\beta$ as the number of instances gets larger (Tsiatis, 1990; Jin et al., 2003). Although these asymptotic results might not be directly generalized to the non-linear predictor function, we expect that hidden layers would be able to assess effective representations $W_i$ with non-linear feature transformation, as evidenced by extensive quantitative studies. Note that, to keep theoretical alignment, it is encouraged to set the last layer as a linear transformation with an output dimension of 1 to mimic the standard linear model following non-linear representation. In addition, a robust estimation against outlying instances can be attained, depending rank of residual terms along with their difference.

### 3.2 Prediction of Survival Quantities

Predicted output $g(X_i; \hat{\theta})$ from trained *DART* model represents estimated expectation of $\log T_i$ conditional on $X_i$, i.e. mean log-transformed survival time with given feature information of $i$th instance. However, estimating survival quantities (e.g. conditional hazard function) cannot be directly done for AFT-based models. Instead, we utilize the Nelson-Aalen estimator (Aalen, 1978), verified to be consistent under the rank-based semiparametric AFT model (Park & Wei, 2003). Define $N(t; \theta) = \sum_{i=1}^{N} N_i(t)$ and $Y(t) = \sum_{i=1}^{N} Y_i(t)$, where $N_i(t) = I(e_i \leq t, \delta_i = 1)$ and $Y_i(t) = I(e_i > t)$ are the counting and the at-risk processes, respectively. Then the Nelson-Aalen estimator of $H_0(t)$ is defined by

$$\hat{H}_0(t) = \int_0^t \frac{I\{Y(u) > 0\}}{Y(u)} dN(u). \tag{8}$$

The resulting conditional hazard function given $X_i$ is defined by

$$\hat{h}(t|X_i) = \hat{h}_0[t \exp\{-g(X_i; \hat{\theta})\}] \exp\{-g(X_i; \hat{\theta})\}, \tag{9}$$

where $\hat{h}_0(\cdot) = d\hat{H}_0(\cdot)$ is pre-trained baseline hazard function using Nelson-Aalen estimator. Consequently, conditional survival function can be estimated by relationship $\hat{S}(t|X_i) = \exp\{-\int_0^t \hat{h}(t|X_i)dt\}$, providing comparable predictions to other time-to-event regression models. In practice, training set is used to get pre-trained Nelson-Aalen estimator.

## 4 EVALUATION CRITERIA

In this section, we evaluate models with two metrics for quantitative comparison: concordance index (CI) and integrated Brier score (IBS).

**Concordance Index.** Concordance of time-to-event regression model represents the proposition: if a target variable of instance $i$ is greater than that of instance $j$, then the predicted outputs of $i$ should be greater than that of $j$. By letting target variable $y$ and predicted outcome $\hat{y}$, concordance probability of survival model can be expressed as $P(\hat{y}_i > \hat{y}_j | y_i > y_j)$, and concordance index measures the probability with trained model for all possible pairs of datasets (Harrell et al., 1982). With non-proportional-hazards survival regression models like *Cox-Time* or Lee et al. (2018), however, Harrell et al. (1982) cannot be used to measure discriminative performance properly. For fair comparison of survival regression models, time-dependent concordance index (Antolini et al., 2005), or $C^{\text{td}}$ was used for those baseline models proposed by Kvamme et al. (2019) to account for tied events. $C^{\text{td}} \in [0, 1]$ can be regarded as AUROC curve for time-to-event regression model, denoting better discriminative performance for a value close to 1. Note that standard concordance index yields identical results with $C^{\text{td}}$ for AFT-based models.

**Integrated Brier Score.** Graf et al. (1999) introduced generalized version of Brier score (Brier, 1950) for survival regression model along with inverse probability censoring weight (IPCW), which can be described as:

$$\text{BS}(t) = \frac{1}{N} \sum_{i=1}^{N} \frac{\hat{S}(t|X_i)^2 I(y_i \leq t, \delta_i = 1)}{\hat{G}(y_i)} + \frac{1}{N} \sum_{i=1}^{N} \frac{(1 - \hat{S}(t|X_i))^2 I(y_i > t)}{\hat{G}(t)} \tag{10}$$

where $\hat{G}(t) = \hat{P}(C > t)$ is a Kaplan-Meier estimator for censoring survival function to assign IPCW. $\text{BS}(t)$ measures both how well calibrated and discriminative is predicted conditional survival function: if a given time point $t$ is greater than $y_i$, then $\hat{S}(t|X_i)$ should be close to 0. Integrated Brier score (IBS) accumulates BS for a certain time grid $[t_1, t_2]$:

$$\text{IBS} = \frac{1}{t_2 - t_1} \int_{t_1}^{t_2} BS(s)ds. \tag{11}$$

If $\hat{S}(t|X_i) = 0.5$ for all instances, then IBS becomes 0.25, thus well-fitted model yields IBS lower than. For experiments, time grids can practically be set to minimum and maximum of $y_i$ of the test set, equally split into 100 time intervals.

## 5 EXPERIMENTS

In this section, we describe our experiment design and results to validate performance of *DART* compared to other time-to-event regression models. Experiments are done with four real-world survival datasets, and baseline models provided by Kvamme et al. (2019) and Chapfuwa et al. (2018), using two evaluation metrics mentioned in previous section.

Table 1: Summary of survival datasets.

| DATASET | SIZE | # FEATURES | % CENSORED |
|---|---|---|---|
| WSDM KKBOX | 2,646,746 | 15 | 0.28 |
| SUPPORT | 8,873 | 14 | 0.32 |
| FLCHAIN | 6,524 | 8 | 0.70 |
| GBSG | 2,232 | 7 | 0.43 |

### 5.1 DATASETS

We choose three common survival datasets and a single large-scale dataset provided by Kvamme et al. (2019). The descriptive statistics are provided in Table 1. First, three common survival datasets that used in this work are the Study to Understand Prognoses Preferences Outcomes and Risks of Treatment (SUPPORT), the Assay of Serum Free Light Chain (FLCHAIN), and the Rotterdam

Table 2: Mean and standard deviation of $C^{\mathrm{td}}$. The **boldface** denotes best performance. PMF denotes a method parameterizing the probability mass function. HAZ and AFT denote hazard-based and AFT-based methods, repectively.

|  | MODEL | WSDM KKBOX | SUPPORT | FLCHAIN | GBSG |
|---|---|---|---|---|---|
| PMF | *DeepHit* | 0.553 (0.002) | 0.645 (0.009) | 0.797(0.015) | 0.684 (0.013) |
| HAZ | *DeepSurv* | 0.841 (0.000) | 0.619 (0.008) | 0.797 (0.013) | 0.685 (0.013) |
|  | *Cox-CC* | 0.836 (0.046) | 0.618 (0.009) | 0.797(0.013) | 0.684 (0.012) |
|  | *Cox-Time* | **0.853** (0.049) | **0.637** (0.009) | **0.800** (0.012) | **0.687** (0.012) |
| AFT | *DRAFT* | 0.861 (0.005) | 0.599 (0.018) | 0.725 (0.057) | 0.611 (0.016) |
|  | *DATE* | 0.852 (0.001) | 0.608 (0.008) | 0.784 (0.009) | 0.598 (0.034) |
|  | *DART (ours)* | **0.867** (0.001) | **0.624** (0.009) | **0.797** (0.014) | **0.687** (0.014) |

tumor bank and German Breast Cancer Study Group (GBSG). In addition, WSDM KKBox from preparation for the 11th ACM International Conference on Web Search and Data Mining is the dataset for customer churn prediction containing millions of instances and 15 covariate variables. With this large-scale dataset, consistency of training procedure and predictive performance would clearly be verified.

## 5.2 BASELINE MODELS

We select six neural network-based time-to-event regression models as our experimental baselines: *DRAFT* and *DATE* (Chapfuwa et al., 2018) as AFT-based models for direct comparison with our model, and *DeepSurv* (Katzman et al., 2018), *Cox-CC* and *Cox-Time* (Kvamme et al., 2019) as hazard-based models, *DeepHit* (Lee et al., 2018) as a PMF-based model for references.

For AFT-based models, *DRAFT* utilizes neural networks to fit log-normal parametric AFT model in non-linear manner. That is, it might be misspecified if true error variable does not follow log-normal distribution. In contrast, *DATE* exploits generative-adversarial networks (GANs) to learn conditional time-to-event distribution and censoring distribution using observed dataset.

In case of hazard-based models, *DeepSurv* fits Cox regression model whose output is estimated from neural networks. The model outperforms the standard CoxPH model in performance, not clearly exceeding other neural network-based models. Furthermore, the proportional hazards assumption still remains unsolved with *DeepSurv*. *Cox-CC* is another neural network-based Cox regression model, using case-control sampling for efficient estimation. While both *DeepSurv* and *Cox-CC* are bounded to proportionality of baseline hazards, *Cox-Time* relieves this restriction using event-time variable to estimate conditional hazard function.

In addition, we include *DeepHit* (Lee et al., 2018) as a reference, which is a survival regression model parameterizing discrete-time hazard rate with neural networks based on survival probability mass function (PMF), considering its contribution to alleviate the fundamental assumption of hazard-based and AFT-based models. Although its prediction performance has been reported prominent to others, the training procedure is quite unstable which is a critical shortcoming for practical application.

Except for neural network-based models, we exclude other machine learning-based models from baselines regarding comparison from previous studies. Some neural network-based models are excluded as well in this study since we focus on alleviating fundamental assumptions such as proportionality and parametric distribution. Note that comparing hazard-based models and AFT-based models has rarely been studied due to their difference in concepts: modeling hazard function and modeling time-to-event variable. Despite models can be evaluated with common metrics, analysis upon numerical experiments has to be cautious, especially between a hazard-based model and an AFT-based model.

Table 3: Mean and standard deviation of Integrated Brier Score (IBS).

|  | MODEL | WSDM KKBOX | SUPPORT | FLCHAIN | GBSG |
|---|---|---|---|---|---|
| PMF | *DeepHit* | 0.124 (0.001) | 0.221 (0.034) | 0.160 (0.081) | 0.183 (0.015) |
| HAZ | *DeepSurv* | 0.111 (0.000) | **0.190** (0.004) | **0.101** (0.006) | **0.174** (0.004) |
|  | *Cox-CC* | 0.115 (0.012) | 0.191 (0.003) | 0.122 (0.028) | 0.177 (0.004) |
|  | *Cox-Time* | **0.107** (0.009) | 0.194 (0.006) | 0.114 (0.016) | **0.174** (0.005) |
| AFT | *DRAFT* | 0.147 (0.002) | 0.314 (0.043) | 0.144 (0.022) | 0.310 (0.010) |
|  | *DATE* | 0.131 (0.002) | 0.227 (0.004) | 0.124 (0.012) | 0.204 (0.004) |
|  | *DART (ours)* | **0.108** (0.001) | **0.176** (0.005) | **0.068** (0.007) | **0.150** (0.023) |

## 5.3 MODEL SPECIFICATION AND OPTIMIZATION PROCEDURE

For a fair comparison, we apply neural network architecture used in Kvamme et al. (2019): MLP with dropout and batch-normalization. Every dense blocks are set to have the equal number of nodes, no output bias is utilized for output layer, and ReLU function is chosen for non-linear activation for all layers. Preprocessing procedure has also been set based on Kvamme et al. (2019) including standardization of numerical features, entity embeddings (Guo & Berkhahn, 2016) for multi-categorical features. The dimension of entity embeddings is set to half size of the number of categories. In addition, due to the fact that parameters of AFT-based models tend to be influenced by scale and location of the target variable, $y$ has been standardized and its mean and variance are separately stored to rescaled outputs. For SGD algorithm, AdamWR (Loshchilov & Hutter, 2017) is used as implemented by Kvamme et al. (2019) with one epoch of an initial cycle length. We also set the cycle length to double after each cycle. The details about data split and hyperparameter search are described in the appendix.

## 5.4 PERFORMANCE EVALUATION

To measure discriminative performance of outputs, we exploit standard C-index (Harrell et al., 1982) for AFT-based models while letting hazard-based models to utilize $C^{\text{td}}$ since equivalent evaluation is possible for AFT-based models including *DART* since it outputs a single scalar value to evaluate ranks. In terms of survival calibration, we implement our own function to obtain IBS based on its definition, due to the fact that evaluation methods of the conditional survival function and IPCW provided by Kvamme et al. (2019) are not compatible with AFT-based models. Specifically, we first fit Kaplan-Meier estimator upon standardized training set, and subsequently evaluate conditional survival estimates and IPCW utilizing estimated residuals, following the definition of baseline hazard function of AFT framework rather than to use time-to-event variable directly. For numerical integration, we follow settings of time grid from Kvamme et al. (2019), and standardize the grid with mean and standard deviation stored with standardization procedure of training set. By doing so, IBS can be compared upon identical timepoints for both hazard-based models and AFT-based models.

## 5.5 SUMMARY OF RESULTS

Experiment results are provided in Table 2 and 3. In summary, *DART* is competitive in both discriminative and calibration performance, especially for large-scale survival datasets. Specifically, *DART* yields consistent results for WSDM KKBox dataset compared to other baselines, maintaining competitive performance in terms of $C^{\text{td}}$ and IBS. We point out that *DART* is the most powerful and AFT-based time-to-event model that can be a prominent alternative when hazard-based models might be not working.

## 6 ANALYSIS

We provide analysis on experimental results, pointing out strengths of *DART* model in terms of performance metrics.

Table 4: Comparison of the training time (seconds) per epoch over the KKBox dataset.

|  | *DeepHit* | *DeepSurv* | *Cox-CC* | *Cox-Time* | *DRAFT* | *DATE* | *DART (ours)* |
|---|---|---|---|---|---|---|---|
| Time | 37.36 | 27.81 | 44.86 | 42.60 | 759.04 | 2024.19 | 29.93 |

**Characteristic of *DART* for large-scale dataset.** As provided in Table 2 and 3, *DART* generally yields prominent survival calibration performance with small variance in terms of IBS. Especially for large-scale dataset (KKBox), *DART* shows state-of-the-art performance with the smallest variance in evaluated metrics. This result comes from the characteristic of rank-based estimation strategy. Specifically, on the basis of asymptotic property of Eq. (7), estimated model parameters get stable and close to true parameter set, when the size of dataset gets larger. Thus, once the trained model attains effective representation ($W_i$ in Eq. (7)) from hidden layers via stochastic optimization methods, *DART* is able to provide stable outputs with strong predictive power, without sophisticated manipulation upon time-to-event distribution.

**Comparison with AFT-based models.** In case of *DRAFT*, model does not generally perform well for both $C^{\text{td}}$ and IBS for most datasets. This is attributed to the fact that *DRAFT* is a simple extension of the parametric AFT model with log-normality assumption. Thus, this approach is quite sensitive to true underlying distribution of dataset. On the other hand, *DATE* yields clearly improved performance against *DRAFT* especially for survival calibration in terms of IBS. Unlike *DRAFT*, *DATE* utilizes GAN to learn conditional error distribution without parametric assumption, allowing the model to yield more precise survival calibration. However, time-to-event distribution is trained with divided loss functions by optimizing two tuning hyperparameters in Eq. (4). This approach can be significantly affected by well-tuned hyperparameters and heavy computation is required to this end, resulting insufficient performance. Meanwhile, as illustrated in Figure 1, *DART* has advantages of simplicity in theoretical and practical points compared to the other AFT-based models.

**Comparison with hazard-based and PMF models.** As previously reported by Kvamme et al. (2019), *Cox-Time* shows competitive performance against other hazard-based models, directly utilizing event-time variable to model conditional hazard function. However, we found out that *Cox-Time* requires precise tuning of additional hyperparameters ($\lambda$ and Log-durations) largely affecting predictive performance. *DeepHit*, as a PMF-based model, yields relatively poor performance in our experiments for most datasets especially in terms of IBS, inconsistent with a previous study. Note that, however, *DeepHit* was originally designed to handle the competing-risks problem, thus evaluation with predictive power might not be comparable. In contrast, *DART* shows smaller variance in evaluation metrics as the size of data increases, ensuring stable output for large-scale dataset with asymptotic property which is crucial for practical application.

**Comparison of the required time for optimizing each model.** To verify the compatibility for large-scale data, we measure the training time of each model. We strictly bound the scope of the target process, as from data input to parameter update excluding other extra steps. Also, all models are evaluated with the same number of nodes, layers, and batch size. All experiments were run on a single NVIDIA Titan XP GPU. Table 4 shows that the simplicity of *DART* leads to practical efficiency, while *DATE* is computationally expensive due to the generator-discriminator architecture.

To summarize, we suggest that *DART* would be a powerful alternative to other time-to-event regression models ensuring stable performance with less time consumption.

## 7 CONCLUSION

In this work, we propose simple time-to-event regression model, namely *DART*, utilizing semiparametric AFT rank-regression method and deep neural networks to alleviate strict assumptions and to attain practical usefulness in terms of high and stable predictive power. Through experiments, our model was shown to be prominent in discriminative and calibration performance even with the large-scale dataset. Although we do not yet cover more complex censoring data, such as competing risks and interval censoring, our approach might be able to provide a stable baseline to handle those tasks in near future with a simple modification of our loss function.

ETHICS STATEMENT

We believe that our model can be widely applied in many fields such as biostatistics and treatment recommendation with contributions to society and human well-being. In our experiments, we only used public datasets without any conflicts of interest and sponsorship. We hope that our model's simplicity and public code bring profits to practitioners all over the place.

REPRODUCIBILITY STATEMENT

We clarify that the reported results are based on multiple repetitions of experiments to remove the outlier effect and our codes are available in public. To obtain reproducible and reliable results, we repeated experiments with the best configuration and various random seeds for all models over KKBox dataset and conducted five-fold cross-validation for small-size datasets. As described in the paper and appendix, the measurements of the training time are under the strict control of the experimental setup with repetitions.

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

# A  APPENDIX

## A.1  ADDITIONAL PARAMETER ESTIMATION METHODS

As extensions, we introduce two additional parameter estimation methods for *DART*: error-based method and likelihood-based methods.

### A.1.1  ERROR-BASED ESTIMATION

While rank-based method (Jin et al., 2003) mainly relies on ranks of residual, Buckley & James (1979) proposed least-squares estimator for right-censored survival dataset, known as Buckley-James estimator. With generality, we set Buckley-James type, or error-based loss function for *DART*:

$$L_{\text{Error}}(\theta) = \left( \frac{1}{N} \sum_{i=1}^{N} | \log \hat{y}_i - g(X_i; \theta) |^p \right)^{1/p}$$

where $p \in \{1, 2\}$ and

$$\hat{y}_i = \delta_i \log y_i + (1 - \delta_i) \left\{ \frac{\int_{e_i}^{\infty} u d\hat{F}_\theta(u)}{1 - \hat{F}_\theta(e_i)} + g(X_i; \theta) \right\}$$

is observed time-to-event variable imputed with conditional expectation of residual.

Specifically, $\int_{e_i}^{\infty} u\hat{F}_\theta(u)/(1 - \hat{F}_\theta(e_i))$ denotes $E(e_i | T > C, X_i)$, expected value of residual for censored instance conditional on given featire information $X_i$, where $\hat{F}_\theta$ is cumulative hazard function that estimated via nonparametric methods like Nelson-Aalen estimator. Thus, this method is equivalent to least-squares estimation where incomplete information of censored instance has been corrected with conditional probability distribution of residual and predicted values. $L_{\text{Error}}$ becomes mean-absolute-error (MAE) objective function for $p = 1$ and root-mean-squared-error (RMSE) objective function for $p = 2$, that are standard form of loss of regression problem.

Error-based estimation provides much realistic predicted values for log of time-to-event variable, while rank-based method stands a chance of underevaluating scale of target variable since rank of residuals controls the loss function rather than magnitude of error. One of the shortcomings of error-based method is, however, conditional expectation of residual is neither continuous nor componentwise monotone in parameter set $\theta$ (Jin et al., 2006), thus optimization through SGD algorithms tends to fluctuating, which is verified through series of experiments. This has to be handled via controlling appropriate weight-decay or learning rate in practical applications, while rank-based methods is relatively robust to those configuration settings.

### A.1.2  LIKELIHOOD-BASED ESTIMATION

Maximum-likelihood estimation (MLE) is the core element of statistical inference, providing both strong theoretical background on estimated parameters and accurate estimator itself. Zeng & Lin (2007) suggested kernel-smoothed version of MLE for semiparametric AFT model, claiming that nonparametric negative log-likelihood function can approximately yield minimum via kernel smoothing. With this work, we designed objective function for MLE of *DART*:

$$
\begin{aligned}
L_{\text{MLE}}(\theta) = & \frac{1}{N} \sum_{i=1}^{N} \delta_i g(X_i; \theta) + \frac{1}{N} \sum_{i=1}^{N} \delta_i e_i \\
& - \frac{1}{N} \sum_{i=1}^{N} {}_i \log \left\{ \frac{1}{n a_n} \sum_{j=1}^{N} \delta_j K \left( \frac{e_j - e_i}{a_n} \right) \right\} \\
& + \frac{1}{N} \sum_{i=1}^{N} \delta_i \log \left\{ \frac{1}{N} \int_{-\infty}^{(e_j - e_i)/a_n} K(s) ds \right\}
\end{aligned}
\tag{12}
$$

where $K(\cdot)$ is zero-mean symmetric kernel function and $a_n$ is a bandwidth parameter. Common choice of $K(\cdot)$ is probability density function of standard normal distribution, while optimal bandwidth parameter $a_n$ for neural network has not been suggested yet.

Table 5: Hyperparameter search space for GBSG, FLCHAIN, and SUPPORT datasets.

| Hyperparameter | Values |
|---|---|
| # Layers | {1, 2, 4} |
| # Nodes per layer | {64, 128, 256, 512} |
| Dropout | {0.1, 0.2, 0.3, 0.4, 0.5, 0.6, 0.7} |
| Weight decay | {0.4, 0.2, 0.1, 0.05, 0.02, 0.01, 0.001} |
| Batch size | {64, 128, 256, 512, 1024} |
| $\alpha$ (DeepHit) | {0.0,0.1,0.2,0.3,0.4,0.5,0.6,0.7,0.8,0.9,1.0} |
| $\sigma$ (DeepHit) | {0.1, 0.25, 0.5, 1, 2.5, 5, 10, 100} |
| Num. durations (DeepHit) | {50, 100, 200, 400} |
| $\lambda$ (CoxTime and CoxCC) | {0.1, 0.01, 0.001, 0.0} |

Table 6: Hyperparameter search space for the WSDM KKBox dataset.

| Hyperparameter | Values |
|---|---|
| # Layers | {4,6,8} |
| # Nodes per layer | {128, 256, 512} |
| Dropout | {0.0, 0.1, 0.5} |

Minimizing $L_{\mathrm{MLE}}(\theta)$ with respect to $\theta$ yields statistically consistent estimator for $\theta$, meaning that it is possible to get better survival calibration compared to rank-based and error-based estimation. This can be a significant merit for the case that needs precise evaluation of patient-dependent survival probability. In spite of those theoretical strengths of likelihood based method, optimization via batched SGD does not guarantee stable convergence due to the fact that values of $\theta$ close to negative infinite can get sufficiently small loss. Preventing divergence needs detailed control of hyperparameters; bandwidth, learning rate, weight decay, etc.

## A.2 EXPERIMENTAL SETTINGS

### A.2.1 DETAILS IN HYPERPARAMETER SEARCH AND OPTIMIZATION

In this section, we describe details on experimental settings including hyperparameter search space and optimization strategy. Our codes are available in public[1].

***DeepSurv, DeepHit, Cox-CC, Cox-Time, DART.*** The PyCox[2] python package provides the training codes for these models. For WSDM KKBox dataset, we repeated experiments 30 times with best configurations provided by Kvamme et al. (2019). Because train/valid/test split of KKBox dataset is fixed, we didn't perform a redundant search procedure. For the other datasets (SUPPORT, FLCHAIN, and GBSG), we performed 5-fold cross-validation as performed at Kvamme et al. (2019) because the size of datasets is relatively small. At each fold, the best configuration was selected among 300 combinations of randomly selected hyperparameters which are summarized in Table 5. As described in the paper, we used AdamWR (Loshchilov & Hutter, 2017) starting with one epoch of an initial cycle and doubling the cycle length after each cycle. The batch size was set to 1024 and the learning rates were found by Smith (2017) as performed at Kvamme et al. (2019). The early stopping was applied with patience of 10 epochs for all models equally.

***DRAFT, DATE.*** The implementation of *DATE*[3] by the authors includes the code of *DRAFT* as well. We utilized their official codes for all datasets. The batch size for KKBox dataset was set to 8192 because the experiments were not feasible with the batch size 1024 due to their training time. The best configurations of *DRAFT* and *DATE* also were founded by grid search with same hyperparameter search space. We repeated experiments 30 times with the best configuration as mentioned above. For the other datasets, as same with other models, we performed 5-fold cross-

---

[1]https://github.com/dart-submission/dart-submission

[2]https://github.com/havakv/pycox

[3]https://github.com/paidamoyo/adversarial_time_to_event

Table 7: Best configurations for WSDM KKBox dataset. $\alpha$ and $\sigma$ are applied to the DeepHit.

| MODEL | # Layers | # Nodes | Dropout | $\alpha$ | $\sigma$ |
|---|---|---|---|---|---|
| *DeepHit* | 6 | 512 | 0.1 | 0.001 | 0.5 |
| *DeepSurv* | 6 | 256 | 0.1 | - | - |
| *Cox-CC* | 6 | 128 | 0.0 | - | - |
| *Cox-Time* | 8 | 256 | 0.0 | - | - |
| *DRAFT* | 8 | 128 | 0.0 | - | - |
| *DATE* | 8 | 512 | 0.5 | - | - |
| *DART (ours)* | 6 | 256 | 0.0 | - | - |

validation and chose the best configuration among 300 random hyperparameter sets at each fold. The early stopping was applied as done in other models equally.

### A.2.2 COMPARISON OF THE TRAINING TIME

For a fair comparison, we measured the training time for only the optimizing phase because there are various extra steps for each model. We strictly set the range of the target process, as from putting data to updating model parameters. The specifications of all models were set equally: the number of nodes 256, the number of layers 6, and the batch size 1024. With the consumed time of 1000 iterations, we calculated the training time for a single epoch. We excluded the first iteration that is an outlier in general. To obtain reliable results, we repeated five times and reported the average values.

### A.2.3 FUTURE WORKS

Even though *DART* alleviates the fundamental and strict assumptions, there are unresolved assumptions yet. We assume the non-informative right-censoring as most time-to-event models did as basic assumptions. Also, AFT models inherently consider features as accelerating or delaying factors of the event time. Addressing these basic assumptions is the direction we should go in order to build a more generalized model.

As mentioned in the paper, some of the researches resolved issues of time-to-event modeling such as competing risks, discrete time, and time-varying covariate effect (Katzman et al., 2018; Lee et al., 2018; Ren et al., 2019; Kvamme & Borgan, 2019; Avati et al., 2020; Tarkhan et al., 2021; Rahman et al., 2021; Su, 2021). Even though we didn't focus on these issues at this work, but one future direction can be to integrate these techniques with *DART* because such assumptions are important as well and *DART* is compatible with others.

We are also aware of a series of researches that use various neural networks such as recurrent neural networks (RNNs), convolutional neural networks (CNNs), variational auto-encoder (VAE), or transformers (Lee et al., 2019; Nagpal et al., 2021; Gao & Cui, 2021; Kim et al., 2020; Kamran & Wiens, 2021; Hu et al., 2021). All of these studies are complementary to our model rather than competing because we can easily replace our $g_\theta$ with RNNs, CNNs, VAE, and transformers.

