# OpenReview forum: "Towards simple time-to-event modeling: optimizing neural networks via rank regression"
_ICLR.cc/2022/Conference — ICLR 2022 Submitted_

### Official Review · Reviewer_qkV5 · 2021-10-22

**Correctness:** 2
**Technical Novelty And Significance:** 1
**Empirical Novelty And Significance:** 2
**Recommendation:** 3
**Confidence:** 4

**Main Review:**

The paper is easy to follow with competitive quantitative results. However, the technical contributions are lacking, there are some misleading statements, and the writing needs improvement. Below are specific examples.

-  Introduction: "The accelerated failure time model (AFT) or accelerated life model relates the logarithm of the failure time linearly to the features." This statement is not necessarily true for parametric AFT other than log-Normal, *e.g.*,  Weibull, Exponential, *, etc.*
-  The $L_{\rm CoxPH}$  (Eq. 2) definition is not correctly specified. Note the risk set $\mathcal{R_i}$ should include censored on non-censored times $T_j$ s.t. $T_j < T_i$
- Sec 3.2: "However, estimating survival quantities (e.g. conditional hazard function) cannot be directly done for AFT-based models." This statement is not necessarily true for some parametric AFT models, *e.g.*, Weibull, Exponential, *, etc.*
- Sec 3.2: While paper claims to make time-to-event  predictions $\hat{T} = \exp(g(X_i, \theta))$ it is not clear why those predictions can not be used to directly estimate $\hat{S}(t| X)$. Instead, the paper proposes a semiparametric conditional hazard transformation similar to CoxPH without providing any justification.
- Sec 4: Is there evidence that supports this statement  "Note that standard concordance index yields identical results with ${\rm C}^{\rm td}$ for AFT-based models "?
- Sec 5.4:  For fair comparisons, the paper should compare AFT and hazard models using a similar metric, either C-Index or ${\rm C}^{td}$ or both.
- For a comprehensive evaluation, the paper should provide additional qualitative results, *e.g.*, model predictions against ground truth, and calibration curves.

**Minor Issues**
-  Missing $\xi \sim F_{\xi}$ in third term (Eq. 4)
- Introduction: Typo ATF should be AFT
- Sec 4: "well-fitted model yields IBS lower than"  sentence is incomplete


**Summary Of The Paper:**

The paper extends the previously proposed linear semiparametric AFT model based on Gehan’s rank statistic to a nonlinear setup. The nonlinear model termed Deep AFT Rank-regression for Time-to-event prediction model (DART) is parameterized by a neural network. Experimental results across four datasets show a competitive advantage over baselines per concordance index (C-Index), integrated Brier score (IBS), and training time.

**Summary Of The Review:**

Apart from empirical competitive quantitative results against baselines, the technical contributions are not clear. The semiparametric AFT model based on Gehan’s rank statistic objective function has been previously proposed. Additionally, while the paper claims to make time-to-event predictions, the conditional hazard transformation of model predictions seems to indicate the model predicts hazards instead.

---

> ### Author Response · Authors · 2021-11-18
> **Response for reviewer qkV5**
>
> We thank the reviewer for the constructive feedback and valuable comments, here we address the concerns.
>
>
> *Q1.  "The accelerated failure time model (AFT) or accelerated life model relates the logarithm of the failure time linearly to the features." This statement is not necessarily true for parametric AFT other than log-Normal, e.g., Weibull, Exponential, , etc.*
> - As commonly understood, AFT framework models logarithm of failure time and related feature variables in linear fashion, regardless of parametric or semi-paramertic approach. Parametric distributional assumptions, like log-normal, Weibull, Exponential, are imposed upon the error term $\epsilon$, so this model is still linear for logarithm of failure time. Please refer to section 2.2 of our paper, or section 2 of [1].
>
>
> *Q2. The LCoxPH (Eq. 2) definition is not correctly specified. Note the risk set Ri  should include censored on non-censored times Tj  s.t. Tj<Ti*
> - We would make our explanations clearer, but the definition of risk set that you had mentioned is included in our work. Please refer to the comment on section 2.1:
> > Letting $R_i$ be the set of all individuals “at risk”, meaning that are not censored and have not experienced the event before $T_i$ , statistically efficient estimator for regression coefficients can be obtained minimizing the loss function with respect to $\beta$
>
>
> *Q3. Sec 3.2: "However, estimating survival quantities (e.g. conditional hazard function) cannot be directly done for AFT-based models." This statement is not necessarily true for some parametric AFT models, e.g., Weibull, Exponential, , etc.*
> - Thank you for your appropriate comment. It is true that direct estimation of survival quantities is possible for standard parametric AFT models including Weibull or Exponential distributions. We will fix our statement clearer, limiting the range for semiparametric approach only.
>
>
> *Q4. Sec 3.2: While paper claims to make time-to-event predictions T^=exp⁡(g(Xi,θ))  it is not clear why those predictions can not be used to directly estimate S^(t|X). Instead, the paper proposes a semiparametric conditional hazard transformation similar to CoxPH without providing any justification*
> - First, it is impossible to directly estimate the survival quantities with semiparametric AFT model, including our proposed method DART. This is because semiparametric approach does not assume distributional assumption unlike parametric AFT models, while estimation of survival function $S(t|X)$ requires specification of conditional probability distribution function $f(t|X)$. Thus, estimation of survival function, which is a function of conditional probability distribution function, cannot directly be done with semiparametric methods.
> This is why we adopted “conditional hazard transformation” to estimate survival function, whose theoretical justification had been provided by [2]. We would add detailed explanations of theoretical justification within the supplementary section.
>
>
> *Q5. Sec 4: Is there evidence that supports this statement "Note that standard concordance index yields identical results with Ctd for AFT-based models "?*
> - Please refer to section 4.1 of [3] for a detailed explanation. In summary, it is equivalent to apply standard C-index and time-dependent C-index ($C^{td}$) for non-time-varying survival regression models.
>
>
> *Q6. Sec 5.4: For fair comparisons, the paper should compare AFT and hazard models using a similar metric, either C-Index or Ctd or both.*
> - As mentioned above, except for time-varying models like CoxTime and DeepHit, C-index and Ctd are measured equivalently so the comparison on this evaluation is fair. If we use the standard C-index for time-varying modesl like CoxTime, the performance would be degraded and it will be an unfair comparison.
>
>
> *Q7. For a comprehensive evaluation, the paper should provide additional qualitative results, e.g., model predictions against ground truth, and calibration curves.*
> - Thanks for pointing this out. We had first planned to add comparison section of prediction of true failure time variable for AFT-based models, but we decided not to due to the limit of length. However, please note that Integrated Brier score (IBS) partially measures the calibratedness of conditional survival curve, which you have mentioned.
>
>
> *Q8. The semiparametric AFT model based on Gehan’s rank statistic objective function has been previously proposed.*
> - Best of our knowledge, Gehan-type semiparametric AFT model has not been explored for optimization of neural networks, in spite of its nice theoretical properties. If you know any related works using this method, please let us know. Also, another thing of our contributions is to introduce Gehan-type estimation technique to the machine learning community, which is surely new for ML researchers. As done in numerous great works, bridging statistically supported techniques and neural networks is not trivial at all. Please consider this point for your decision.

---

> > ### Author Response · Authors · 2021-11-18
> > **Response for reviewer qkV5**
> >
> >
> > *Q9. Additionally, while the paper claims to make time-to-event predictions, the conditional hazard transformation of model predictions seems to indicate the model predicts hazards instead.*
> > - Basically, DART is an AFT model that directly predicts time-to-event variables. As we mentioned, we carefully address this concern for comparison to other hazard-based baselines. We show the conditional hazard estimation for a fair comparison and more plentiful content for readers but the main objective of DART is time-to-event prediction.
> >
> >
> > We also checked the minor issues and will correct them in the revised version.
> >
> > [1] Wei, L. J. (1992). The accelerated failure time model: a useful alternative to the Cox regression model in survival analysis. Statistics in medicine, 11(14‐15), 1871-1879.
> >
> > [2] Park, Y., & Wei, L. J. (2003). Estimating subject‐specific survival functions under the accelerated failure time model. Biometrika, 90(3), 717-723.
> >
> > [3] Håvard Kvamme, Ørnulf Borgan, Ida Scheel. Time-to-Event Prediction with Neural Networks and Cox Regression. JMLR 2019.

---

### Official Review · Reviewer_7ymN · 2021-10-27

**Correctness:** 2
**Technical Novelty And Significance:** 2
**Empirical Novelty And Significance:** 2
**Recommendation:** 1
**Confidence:** 5

**Main Review:**

I found this paper straightforward to read although I have several large concerns.

My first concern is that the exposition is overly complicated even though the key idea of the paper is extremely simple. The proposed model DART can be derived by just taking the known rank estimator by Jin et al (2003) and swapping out the inner product in their equation (2.1) with a neural net, and then just train using minibatch gradient descent instead of linear programming. In other words, much like how Faraggi and Simon (1995) swapped out the inner product in the semiparametric Cox model to be a neural net to come up with the same neural net model as DeepSurv (Katzman et al 2018), this paper does the same thing with the already existing semiparametric AFT model (which already was known before the Jin et al (2003) paper but the optimization procedures in estimating the regression coefficients were not optimal). From a technical novelty/innovation perspective, there is very little that is conceptually new. Instead, the paper is motivated in a way that spends way too much emphasis/text on describing standard results from survival analysis (e.g., what is a Cox model, what is an AFT model, why one would use standard Cox PH vs AFT, that the Cox model can be made time-dependent, that AFT models can be specified in both parametric or semiparametric forms, etc). I'd suggest perhaps having your background instead focus on existing semiparametric AFT literature, that you're simply doing a straightforward neural net extension (replace inner product with neural net), and comparing your model with DRAFT and DATE (I found Figure 1 very helpful). I'd suggest reducing the amount of text spent on explaining what hazards models are or why one should use hazards models over AFT models as this is a very old debate at this point (from what I can tell it really just depends on the data). Of course, a Weibull regression model is both a hazards model and an AFT model.

My next concern is that the experimental results are inconsistent with existing literature. As is, the experimental results presented make it seem that DART is not a clear winner over top-performing baselines. However, once we consider that some of the numbers are quite off from literature, how good DART is seems even more suspect. For example, your reported DeepHit $C^{\text{td}}$ number for the KKBox dataset is dramatically off from Kvamme et al (2019)---basically according to what they get, DeepHit gets 0.888 which is higher than DART 0.867. Some of your IBS numbers across methods also seem off. Some explanation of these discrepancies would be helpful.

There is another exposition issue: early on in the paper, the text makes it seem like the Cox model does not make a linear assumption whereas AFT does. This isn't true. Cox assumes the partial log likelihood is linear. AFT assumes the log survival time is linear. In other words, both make linearity assumptions, just for different quantities.

References:
- David Faraggi and Richard Simon. A neural network model for survival data. Statistics in Medicine 1995.
- Zhezhen Jin, D. Y. Lin, L. J. Wei, Zhiliang Ying. Rank-based inference for the accelerated failure time model. Biometrika 2003.
- Jared L. Katzman, Uri Shaham, Alexander Cloninger, Jonathan Bates, Tingting Jiang, and Yuval Kluger. DeepSurv: personalized treatment recommender system using a Cox proportional hazards deep neural network. BMC Medical Research Methodology 2018.
- Håvard Kvamme, Ørnulf Borgan, Ida Scheel. Time-to-Event Prediction with Neural Networks and Cox Regression. JMLR 2019.

**Summary Of The Paper:**

This paper proposes a (deep) neural net extension to an existing ranking-based estimator used for the semiparametric AFT model (Jin et al 2003). The resulting method achieves time-dependent concordance index and integrated Brier score values that are competitive compared to baseline deep AFT models.

**Summary Of The Review:**

This paper has extremely limited technical novelty, spends too much text on explaining existing standard results/definitions from survival analysis, and has experimental results that are suspect (inconsistent with existing literature).

---

> ### Author Response · Authors · 2021-11-18
> **Response for reviewer 7ymN**
>
> We thank the reviewer for the valuable comments, here we address the concerns. We believe you will change your decision if you read them carefully.
>
> *Q1. My first concern is that the exposition is overly complicated even though the key idea of the paper is extremely simple. $\cdots$ From a technical novelty/innovation perspective, there is very little that is conceptually new.*
> - It is true that we adopted a similar strategy as [1] and [2] previously did for their work. However, one of the key aspects of our paper is introducing a new methodology whose usefulness has been proven in the statistics community like the aforementioned works. We are quite sure that most readers and researchers in the representation learning community would be unfamiliar with concepts of AFT and semiparametric approach in spite of their strong theoretical background and nice properties, which can be much improved combined with current state-of-the-art representation learning methods in terms of practical applications. Especially, Gehan-type rank loss function, as an optimization strategy for a survival regression model, has not been introduced to ML community. We respectfully ask you to consider this point.
> - Plus, under standard regularity conditions like boundedness of probability distribution function or differentiability, Gehan-type rank loss function is equivalent to objective function, which becomes zero when the estimated values of model parameters are equal to true unknown values [3]. Although it is not guaranteed that numerical optimization is always valid with non-linear transformation of feature variables, this might be the common limit of representation learning dealing with non-convex objective function. As provided in experimental results, however, optimization results for DART are shown to be consistent for large-scaled dataset (KKBox), yielding smaller variance in evaluated metrics.
>
>
> *Q2. Instead, the paper is motivated in a way that spends way too much emphasis/text on describing standard results from survival analysis $\cdots$ I'd suggest reducing the amount of text spent on explaining what hazards models are or why one should use hazards models over AFT models as this is a very old debate at this point (from what I can tell it really just depends on the data).*
> - Thank you for your suggestion. Our intention is to point out the usefulness of combining statistical approaches with nice properties and powerful representation learning techniques. We thus have assigned a considerably large part of entire paper to deliver detailed backgrounds of our approach, for both readers from statistical and machine-learning communities. Accepting your suggestion, we would consider reducing the explanation parts.
>
>
> *Q3. My next concern is that the experimental results are inconsistent with existing literature. $\cdots$ Some explanation of these discrepancies would be helpful.*
> - We did entire experiment upon baseline models by ourselves rather than borrowing pre-reported scores for fair comparison, exploiting [4] and [5] to fit DATE/DRAFT models and other hazard-based models, respectively. Especially, in case of KKBox dataset, we used exactly the same hyperparameter sets previously reported by the authors. All experiments were repeated 30 times under the same settings not to be affected by outlying results, and we are willing to open all records and codes through our Wandb dashboard after blind-review process. Although we did all of our efforts to get fair results for baseline methods, reproducibility issue of implemented codes cannot be resolved. We will add explanations upon results aligning with supplementary contents.
> - Plus, in spite of crossing evaluation metrics in some datasets, DART performed best against other baseline models for KKBox dataset, which is a large-scaled survival dataset with more than millions of instances, attaining smallest variance in evaluation metrics. This fact implies that our model DART is much competitive for real-world huge biomedical data, without requiring complex settings like hyperparameters for loss function. Please consider those points to review on our experimental results.
>
>
> *Q4. There is another exposition issue: early on in the paper, the text makes it seem like the Cox model does not make a linear assumption whereas AFT does. This isn't true. Cox assumes the partial log likelihood is linear. AFT assumes the log survival time is linear. In other words, both make linearity assumptions, just for different quantities.*
> - It is unfortunate to mention, but we did not describe that standard Cox model does not require linearity assumption. Please check section 2.1 which is specifying standard Cox regression model, which is a linear one, and we would be pleased that you point out the part causing such misleading.

---

> > ### Author Response · Authors · 2021-11-18
> > **References**
> >
> > [1] David Faraggi and Richard Simon. A neural network model for survival data. Statistics in Medicine 1995.
> >
> > [2] Jared L. Katzman, Uri Shaham, Alexander Cloninger, Jonathan Bates, Tingting Jiang, and Yuval Kluger. DeepSurv: personalized treatment recommender system using a Cox proportional hazards deep neural network. BMC Medical Research Methodology 2018.
> >
> > [3] Zhezhen Jin, D. Y. Lin, L. J. Wei, Zhiliang Ying. Rank-based inference for the accelerated failure time model. Biometrika 2003.
> >
> > [4] https://github.com/paidamoyo/adversarialtimetoevent
> >
> > [5] https://github.com/havakv/pycox

---

> > ### Comment · Reviewer_7ymN · 2021-11-30
> > **significant exposition changes, also I still find the experiments suspect**
> >
> > Thanks for responding to my review. I think the exposition changes made are a bit substantial and warrants another round of careful review.
> >
> > Regarding the experiments (Q3 in your response): using similar or even much fewer number of nodes and layers for the KKbox dataset for DeepHit (compared to Kvamme et al) I am able to get concordance indices on KKBox that are as good as or *better than* what Kvamme et al. report. That you're getting significantly worse results than them comes off to me as there being a serious bug in your implementation. I completely agree with the idea that you should re-run the baselines yourself rather than just copy over the numbers reported in their paper. I'm just saying that I don't buy that DeepHit does as poorly as you're claiming it to for KKBox.

---

### Official Review · Reviewer_6g2w · 2021-10-30

**Correctness:** 3
**Technical Novelty And Significance:** 2
**Empirical Novelty And Significance:** 1
**Recommendation:** 5
**Confidence:** 2

**Main Review:**

Strength:

1. The idea of combining Gehan’s model with non-linear deep learning model is new
2. Performance is stable on several benchmarks.
3. Clear writing

Weaknesses:
1. The intuition of the approach is not quite clear. Does the proposed model focus on reducing computational time or improving the performance, or both?
2. The results do not support the conclusion. The improvements seem not significant compared to baselines.
3. The methodology parts mostly combine another existing model idea into the current AFT model. Hence, the innovation is limited methodological-wise.

**Summary Of The Paper:**

This paper proposes to combine the idea of Gehan’s rank statistic idea on fitting the AFT model, as well as using the deep learning model as a non-linear method for replacing the linear method in the original AFT model. The authors argue they connect Gehan's non-parametric technique with deep learning models. Experiments on various benchmark datasets show that the proposed model is competitive to state-of-the-art baselines.

**Summary Of The Review:**

While the idea of combining Gehan model with deep learning is new, the results show that the proposed model does not outperform SOTA systems on several benchmarks.

---

> ### Author Response · Authors · 2021-11-18
> **Response for reviewer 6g2w**
>
> Thank you for your time. We have concerns that you have missed some important information in our paper. We prudently provide point-to-point responses to address your concerns.
>
> *Q1. The intuition of the approach is not quite clear. Does the proposed model focus on reducing computational time or improving the performance, or both?*
> - The most attractive aspect of DART is that neural networks can be trained with a simple loss function without predefined parametric distribution or additional hyperparameters for loss function. The traditional AFT models assumed parametric distribution resulting inconsistent performance when data do not follow the predefined distribution. DATE alleviates it via GAN to implicitly learn the distribution but the generator-discriminator architecture is too complicated, and requires hyperparameters to leverage penalty terms. Our goal is to build a neural networks-based AFT model that requires no additional handling, so we exploit the objective function derived from Gehan’s rank statistic which can be used as a simple form. As a result, DART achieves both consistently outstanding performance and faster optimization time compared to other baselines.
>
>
> *Q2. The results do not support the conclusion. The improvements seem not significant compared to baselines.*
> - As we mentioned in the manuscript, Hazard-based models are just references. *Note that comparing hazard-based models and AFT-based models has rarely been studied due to their difference in concepts: modeling hazard function and modeling time-to-event variable.* Therefore, We have to compare DART with AFT-based models as done in [1]. Regarding the qualitative comparison between AFT-based models, as listed below, DART shows significantly outstanding performance in terms of both $C^{td}$ and IBS. Also, even regarding comparison with hazard-based models, DART shows the smaller variance in evaluation metrics as the size of data increases, ensuring stable and consistent performance for the large-scale dataset which is crucial for practical application.
>
> |||||||||||||
> |-----------||:---------||:--------||:---------||:---------||:---------||
> | **Dataset**         | \| **DRAFT**||  \| ||      **DATE**||    \| ||**DART**||
> |                     | \|  *$C^{td}$*|*IBS*|  \| ||    *$C^{td}$* |*IBS*|      \| ||   *$C^{td}$*|*IBS*|
> |KKBox | \| 0.861|0.147|          \| ||    0.852|0.131|              \| || **0.867** | **0.108**|
> |SUPPORT    | \|  0.599|0.314|         \| ||   0.608 |0.227 |              \| || **0.624**  | **0.176**|
> |FLCHAIN | \|  0.725|0.144|         \| ||  0.784 |0.124 |              \| || **0.797**  |**0.068**|
> |GBSG    | \|  0.611|0.310|         \| ||   0.598|0.204 |              \| || **0.687**  | **0.150**|
>
>
> *Q3. The methodology parts mostly combine another existing model idea into the current AFT model. Hence, the innovation is limited methodological-wise.*
> - As done in plentiful nice works, bridging statistically supported techniques and neural networks is not trivial at all. In this work, we show that Gehan-type statistic can lead to the efficient and effective objective function to optimize neural networks through a series of experiments. DATE [1], conversely, shows the neural network (GAN) can replace a part of the statistical models (AFT) by learning implicit distribution. We hope that better models will come out through richer connections between statistics and representation learning communities.
>
> [1] Adversarial Time-to-Event Modeling, Chapfuwa et al., ICML 2018

---

### Official Review · Reviewer_Rqgw · 2021-11-01

**Correctness:** 3
**Technical Novelty And Significance:** 3
**Empirical Novelty And Significance:** 3
**Recommendation:** 6
**Confidence:** 4

**Main Review:**

Strengths:
 - Use of a (for the machine learning literature) novel loss function, based on Gehan’s rank statistic
 - The paper is mostly clearly written and easy to follow
 - The authors put their method into context well


Weaknesses:
 - Due to semi-parametric nature, prediction only possible up until the last observed patient time.
 - The loss function in Jin et al (2003) is derived for the linear case. While the authors argue for how it could work in the non-linear case, there is no guarantee that the optimal set of parameters corresponds to a zero rank statistic in the non-linear case
 - The loss function is quadratic in the training samples, which could lead to bad performance for large data-sets. The authors however show very fast performance on a large data-set, which makes this a minor point.
 - The authors introduce both a neural network architecture for AFT models and an L1 ranking loss. It is not entirely clear how much influence either of these have on the benchmark results. In the appendix the authors introduce other loss functions, but do not include the trained models based on those losses in the benchmark results. The addition of these in table 2 and 3 would help disentangle the influence of the model architecture and of the loss, which would be interesting.

Other:
 - The loss function is quadratic in the number of patients/intervals, whereas the full likelihood is linear. Given this, the speed on KKBOX seems very surprising and is probably due to a small constant. It would be interesting to see the scaling with the number of patients N with real data with both the ranking and the maximum likelihood loss.
 - The authors could make the connection between $\epsilon$ in equation (3) and $h_0$ in (9) more clear as a step before using the Nelson-Aalen estimator to use a semi-parametric AFT.
 - The authors use “calibrated” when talking about Brier score. However traditionally when talking, especially about AFT models, calibration refers to the difference between real and predicted event times and the Brier score is a mix of calibration and classification. The Brier score is the calibration of the survival distribution over time, so calling it calibration is not incorrect, however within the context of the survival literature somewhat misleading. More specific language would make the paper more readable.
 - An introduction or motivation why the loss function is a good loss function would be beneficial for the understanding of the paper. The authors could elaborate on rank statistics and why they are expected to be more stable. Especially, as e.g. DeepHit talks about ranking when talking about concordance, it would be helpful for reading that ranking here is in reference to a weighted log-rank estimating function.
 - It would be interesting to see how well the model does in calibration of the event times (also with the other loss functions), as this is clinically meaningful.
 - The bold facing in table 2 is not consistent (DeepHit should be bold in one case)
 - In the appendix the authors make it seem that there are no convergence guarantees for MLE based estimators. However, also considering the application to AFT models as a special case, there do exist estimators that have some theoretical convergence guarantees (Tang et al 2020, Survival Analysis via Ordinary Differential Equations). A small discussion thereof could be beneficial.

**Summary Of The Paper:**

The authors suggest a neural network based accelerated failure time (AFT) model, as well as an L1-type rank loss, which they argue results in an easy and fast to train model, which is state of the art in terms of two standard evaluation metrics, concordance and integrated Brier score. Following the semi-parametric AFT literature, the baseline hazard is estimated from the data and the full hazard function is parametrized in a way that the parametric part corresponds to a prediction of the failure time. The main innovation of the paper is the use of a novel loss function based on a ranking loss using Gehan’s rank statistic, which while being used in the statistical literature, has not been used in a machine learning setting.

**Summary Of The Review:**

While there is a large number of machine learning methods for survival, by considering a (for the machine learning literature) novel loss function in the AFT setting, this paper can be seen as a meaningful contribution to the machine learning for survival literature. The theoretical foundation of the loss function is however not perfect and the influence of loss function and architecture are hard to disentangle. Due to the form of the loss function, a formal scaling analysis would also be needed. I would recommend a weak accept. I could be convinced to improve of the score with a more convincing argument why the loss term in the non-linear setting is a good loss term, maybe through some upper bound on the loss term.

---

> ### Author Response · Authors · 2021-11-18
> **Response for reviewer Rqgw**
>
> Thank you for your time, we are very glad that you have a positive impression on our work. According to your concerns, we provide the following responses:
>
> *Q1. Due to semi-parametric nature, prediction only possible up until the last observed patient time.*
> - Yes. This is due to the conceptual difference between parametric and semiparametric models. While parametric models are able to predict using pre-defined distribution for the time period after the last observed time, semi-parametric models are limited in this point. However, parametric models should be verified if the data follow the pre-defined distribution or not. When the data do not fit on the parametric distribution, the prediction for the time point after the last observed time is not reliable as well. In the real world, it is hard to find a perfect parametric distribution for event time data so semiparametric models have benefits in this point.
>
>
> *Q2. The loss function in Jin et al (2003) is derived for the linear case. While the authors argue for how it could work in the non-linear case, there is no guarantee that the optimal set of parameters corresponds to a zero rank statistic in the non-linear case.*
> - As we mentioned, (7) is not directly generalized for the non-linear case, so we decomposed neural networks into representation learning part $W$ and a linear layer $\beta$, where $W$ is the feature vector transformed by neural networks with parameter set $\phi$. We expect that the optimal representation feature $W$ can be obtained based on Universal Approximation Theorem. After getting an effective $W$, we can keep theoretical consistency with (7) by setting a last linear layer as $\beta$.
> We also argue that, despite theoretical consistency cannot be verified by now, empirical convergence has been shown through experimental results. In case of KKBox dataset, DART yields lowest variance in evaluated metric scores which means consistent convergence of model parameters $W$ and $\beta$. Considering the implication of asymptotic convergence of Gehan-type rank loss function, assuring convergence of linear parameter $\beta$ for large-scale dataset, we would like to say empirical optimization is valid for DART in case of non-linear setting. The theoretical proof in detail for the non-linear setting will be our future direction.
>
> *Q3. The loss function is quadratic in the training samples, which could lead to bad performance for large data-sets. The authors however show very fast performance on a large data-set, which makes this a minor point.*
> - We trained our model with a simple objective function by constructing sample pairs within each batch. It requires a quadratic of the batch size, not the full training sample size. Therefore, we can make it consistently fast in the large-scale datasets as well. Also, you can find some of the baselines (e.g. CoxTime) that require quadratic computation of the loss. Our simple loss form is efficient in that we perform log-transform once while CoxTime needs two steps of monotonic transforms.
>
> *Q4. The authors introduce both a neural network architecture for AFT models and an L1 ranking loss. It is not entirely clear how much influence either of these have on the benchmark results. In the appendix the authors introduce other loss functions, but do not include the trained models based on those losses in the benchmark results. The addition of these in table 2 and 3 would help disentangle the influence of the model architecture and of the loss, which would be interesting.*
> - Thanks for the suggestion. The comparison between DRAFT and DART can show the influence of the loss function because DRAFT is a simple extension of the AFT model with a neural network that is the same model architecture as ours. DART has a simple loss function form without distribution assumption because we exploit the order of residuals from Gehan’s rank statistic. However, there are several loss functions from semiparametric AFT frameworks as we described in supplementary. The addition of these in the qualitative analysis would be interesting as you suggested so we will add it in the revised version.
>
>
> We appreciate again for your helpful comments so that we can improve our manuscript with the above points and minor concerns mentioned.

---

### Official Review · Reviewer_PtQZ · 2021-11-03

**Correctness:** 2
**Technical Novelty And Significance:** 2
**Empirical Novelty And Significance:** 1
**Recommendation:** 3
**Confidence:** 4

**Main Review:**

This is a somewhat confusing paper for several reasons: i) the problem is set as AFT, however, the likelihood (encoded in the residuals), which is key to AFT is not used for optimization, so one could argue that the proposed approach is not an AFT formulation but rather rank regression with censoring; ii) the rank-based loss function in (6) that is very similar to that in Raykar et al. (NeurIPS 2007: On Ranking in Survival Analysis: Bounds on the Concordance Index), which is used in DRAFT but not discussed in the paper; iii) the experiments show that the proposed model is no better than the standard, linear, Cox proportional hazard in terms of C-index which is the most appropriate performance metric considering the proposed model is optimizing for ordering using a rank statistic; iv) the theoretical results for (7) in the linear case, as the authors discuss are not directly generalizable to nonlinear models.

The proposed approach seems to outperform Cox in large datasets. The authors may consider presenting results on additional large datasets to confirm that it is indeed the case, which in turn will make the experiments section stronger.

Note that DRAFT in Figure 1 is not introduced or discussed in the paper.

**Summary Of The Paper:**

The authors introduce a deep-learning-based extension to accelerated failure time modeling for survival analysis and introduce a rank-regression-type loss function based on Gehan's rank statistic.

**Summary Of The Review:**

The proposed approach presented as a deep-learning extension of AFT optimizes a rank-based loss function very similar to that in Raykar et al. (NeurIPS 2007: On Ranking in Survival Analysis: Bounds on the Concordance Index). The experiments results show that the proposed approach does not outperform standard Cox in 3 out of 4 datasets.

---

> ### Author Response · Authors · 2021-11-18
> **Response for reviewer PtQZ**
>
> Thank you very much for taking the time to review our submission and the comments! We are happy to respond to your comments as below:
>
> *Q1. The likelihood (encoded in the residuals), which is key to AFT is not used for optimization, so one could argue that the proposed approach is not an AFT formulation but rather rank regression with censoring.*
> -  As mentioned in our manuscript with reference to Jin et al. [1], DART is strictly categorized into AFT framework in that AFT models are about the relationship between time-to-event variables and feature variables. The semiparametric AFT framework, as you mentioned, has the strength that it does not require parametric distributional likelihood for optimization. Rank regression with censoring itself is a way of optimization strategy like parametric likelihood, and they are not splitted concepts but are both AFT methods. It might be an unfamiliar concept because it does not use the likelihood function from common distribution, but it is definitely an AFT framework.
>
>
> *Q2.  The rank-based loss function in (6) that is very similar to that in Raykar et al. (NeurIPS 2007: On Ranking in Survival Analysis: Bounds on the Concordance Index), which is used in DRAFT.*
> - In our context, ‘rank’ denotes the **order of residual terms** while DRAFT is optimized by the **order of event time**. Our objective function is derived from Gehan’s rank statistic (in Eq (7)) whose value becomes zero when estimates of model parameters are equivalent to true unknown parameter values, but the objective function of DRAFT is from a lower bound on the concordance index as you mentioned. Even though the two formulas (ours and DRAFT’s) may appear similar, but the implications **are completely different**. We will add an explanation for a comparison of this point.
>
>
> *Q3. The experiments show that the proposed model is no better than the standard, linear, Cox proportional hazard in terms of C-index which is the most appropriate performance metric considering the proposed model is optimizing for ordering using a rank statistic.*
> - As we mentioned above, DART is optimized **by the order of residuals, not the order of event time**. Since the residual is an error term remaining after removing the impact of the feature from the time-to-event variable, it is not a method to directly optimize the concordance of the time-to-event. So, C-index is not the most appropriate performance for DART but it still shows comparable performance on this metric during outperforming others on the IBS metric.
> (+ we didn’t include the standard linear CoxPH model in baselines in this work, but DART clearly outperforms it in our observation.)
>
>
> *Q4. The theoretical results for (7) in the linear case, as the authors discuss are not directly generalizable to nonlinear models.*
> - As we mentioned, (7) is not directly generalized for the non-linear case, so we decomposed neural networks into representation learning part $W_i$ and a linear layer $\beta$, where $W_i$ is the feature vector transformed by neural networks with parameter set $\phi$. We expect that the optimal representation feature $W$ can be obtained based on Universal Approximation Theorem. After getting an effective $W$, we can keep theoretical consistency with (7) by setting a last linear layer as $\beta$.
>
>
> *Q5. The proposed approach seems to outperform Cox in large datasets. The authors may consider presenting results on additional large datasets to confirm that it is indeed the case, which in turn will make the experiments section stronger.*
> - Thanks for the comment. We carefully selected datasets in terms of fair comparison (datasets that are used in baselines), but we agree that additional large-scale datasets would be helpful to show our strength.
>
>
> *Q6. Note that DRAFT in Figure 1 is not introduced or discussed in the paper.*
> - We mentioned DRAFT in the Figure 1 caption, Section 5.2, and Section 6. Especially, Figure 1 provides a clear introduction and comparison of each AFT model. However, we agree that the detailed introduction for DRAFT is helpful to explain the concept and eventually to accentuate the strength of DART. Including all of the above comments, we will add the part for DRAFT.
>
> > (Figure 1 caption) To alleviate the parametric distribution assumption, which DRAFT has...
>
> > (Section 5.2) DRAFT utilizes neural networks to fit log-normal parametric AFT model
> in non-linear manner. That is, it might be misspecified if true error variable does not follow lognormal distribution.
>
> >  (Section 6) This is attributed to the fact that DRAFT is a simple extension of the parametric AFT model with lognormality assumption.
>
>
> We hope all of your concerns will be addressed in this discussion.
>
> [1] Zhezhen Jin, DY Lin, LJ Wei, and Zhiliang Ying. Rank-based inference for the accelerated failure time model. Biometrika, 90(2):341–353, 2003.

---

### Author Response · Authors · 2021-11-30
**Authors' comment**

We appreciate all reviewers’ great effort and time in reviewing our work and also providing constructive feedback.
Unfortunately, we couldn't receive feedback from any reviewer after our rebuttal response, however, we believe that it helps reviewers to understand our work clearly regarding our response to all concerns.

---

### Decision · Program_Chairs · 2022-01-20

**Decision:**

Reject

**Comment:**

Most reviewers came to the conclusion, that this work lacks novelty and theoretical depth. Further severe concerns about the validity of some statements and about the experimental setup have been raised. The rebuttal was not perceived as being fully convincing, and nobody wanted to champion this paper.
I share most of these points of criticism. Although there is certainly some potential in this work, I think it is not ready for publication and would (at least) need a major revision.